# Modern Therapeutic Approaches in Anaplastic Thyroid Cancer: A Meta-Analytic Review of Randomised and Single Arm Studies on Efficacy and Survival

**DOI:** 10.3390/cancers17050777

**Published:** 2025-02-24

**Authors:** Mutahar A. Tunio, Donna Hinder, Blaise Emery, Muhammad H. Riaz, Yusef A. Ibraheem, Krishnendu Kumar Nayak, Wael Mohamed

**Affiliations:** 1South West Wales Cancer Center, Swansea Bay University Health Board, Singleton Hospital, Swansea SA2 8QA, UKwael.mohamed@wales.nhs.uk (W.M.); 2Department of Medicine, Swansea Bay University Health Board, Swansea SA2 8QA, UK; 3School of Medicine, Swansea Bay University, Swansea SA2 8QA, UK

**Keywords:** anaplastic thyroid cancer, mutational/anti-angiogenesis-directed therapies, meta-analysis

## Abstract

The present meta-analysis aimed at the efficacy and safety of modern therapeutic approaches, including targeted therapies, in locally advanced, recurrent and metastatic anaplastic thyroid cancer (ATC). Analyzing 47 studies with 980 patients, the pooled overall response rate (ORR) was 29.7%, with a pooled median overall survival (OS) of 7.2 months and pooled median progression-free survival (PFS) of 5.4 months. Dabrafenib/trametinib (DT), with or without pembrolizumab and lenvatinib plus pembrolizumab (LP), demonstrated superior ORR, PFS and overall survival (OS). DT was identified as a promising first-line treatment for BRAFV600-mutated ATC, while LP showed potential in BRAFV600 wild-type and PDL1-overexpressing cases. Nearly half of the studies reported bio-marker analyses necessitating routine next-generation sequencing (NGS) testing (e.g., BRAFV600, PDL1, RAS). These findings underscore the role of biomarker-driven strategies in optimising ATC treatment.

## 1. Introduction

Anaplastic thyroid cancer (ATC) is a highly aggressive and rare cancer, as defined by the European Union Rare Cancer Surveillance Program (RARECARE), with an incidence beneath six new cases per 100,000 person-years [1]. The majority of ATC patients are diagnosed at an advanced stage (IVB and IVC) associated with compressive symptoms, local invasion and distant metastasis, and only 10% are surgical candidates (IVA) [2,3]. Due to ATC’s ultimate nature, the median survival time is 3–4 months, and the 1-year survival rate is approximately 20% [4].

Due to its rarity, ATC has never been entrant in a large randomised remedial trial. Thus, available data have been primarily obtained from single-centre retrospective studies or phase II non-randomised trials [5]. Historically, doxorubicin monotherapy or a combination of two or more drugs, such as paclitaxel; cisplatin; or carboplatin, doxorubicin and docetaxel, has been considered the most effective treatment. Response rates range from 20% to 50% and significant toxicity [6]. The shift from conventional chemotherapy to precise, molecularly targeted therapies is a rapidly developing phenomenon. Tyrosine kinase inhibitors (TKIs) (lenvatinib, sorafenib, pazopanib) have been trialled since 2008. Lenvatinib has shown meaningful antitumor activity but limited clinical efficacy against ATC [7].

Genomic alterations are highly prevalent in ATC compared to differentiated thyroid cancer [8]. Studies indicate that 95.8% of ATC cases harbour at least one genetic alteration in the MAP kinase pathway or the PIK3/Akt pathway [8,9,10]. The most common mutation is BRAF V600E, observed in 8–87% of cases [8]. Personalised medicine, integrating genomic information, depends on next-generation sequencing (NGS) biomarker assays, which have significantly advanced in ATC. These assays facilitate the identification of targetable mutations and provide timely access to mutationally directed therapies (BRAF V600E mutations with dabrafenib plus trametinib [8]; TRK fusion alterations with entrectinib or larotrectinib [9] and RET fusion with selpercatinib [10]). Despite being based on low-powered studies, these findings have led to the approval of these treatments for ATC. In addition, several small studies of IO alone or combined with TKIs have yielded favourable response and progression-free survival (PFS) rates [11].

Despite current recommendations advocating using NGS and biomarker assays, global clinical practice for ATC management remains highly variable. These inconsistencies in implementation and access have highlighted the need for further evaluation, leading to our present meta-analysis. We aimed to clarify the efficacy and safety of mutational/anti-angiogenesis-directed therapies and contemporary therapies (chemotherapy, radiation therapy, chemoradiation), irrespective of their sequence, to provide some guidance on choosing the most effective regimen for locally advanced, recurrent and metastatic ATC.

## 2. Materials and Methods

We followed PRISMA (preferred reporting items for systemic reviews and meta-analyses) and MOOSE (meta-analyses of observational studies in epidemiology) recommendations to conduct and report present systematic review and meta-analysis [12,13]. The protocol has not been registered.

### 2.1. Intervention

This systematic review and meta-analysis was structured to evaluate the efficacy and survival outcomes of various mutational directed and conventional therapies in patients with unresectable, recurrent and metastatic anaplastic thyroid cancer, compared with tyrosine kinase inhibitors (TKI) [lenvatinib, sorafenib, pazopanib +/− RT, gefitinib, apatinib, sunitinib, imatinib], fosbretabulin (formerly combretastatin A4 phosphate), dabrafenib-trametinib (DT) (BRAF/MEK inhibitor), vemurafenib (*BRAF* inhibitor), immunotherapy (nivolumab, pembrolizumab, nivolumab–ipilimumab, spartalizumab) +/− lenvatinib +/− RT, everolimus (*mTOR* inhibitor), *TRK* inhibitors (larotrectinib, entrectinib), *RET* inhibitor (selpercatinib) and chemotherapy with paclitaxel or carboplatin +/− RT.

### 2.2. Inclusion and Exclusion Criteria

Inclusion criteria were (a) studies on patients with unresectable locally advanced, recurrent or metastatic ATC who had (b) received any of the treatment options as mentioned earlier in any sequence in (c) clinical trials, in either a controlled or single-arm setting, including prospective and retrospective observational studies available as complete publication, with (d) reported efficacy, toxicity and survival outcomes.

Exclusion criteria were (a) studies whose data were erratic for analysis, (b) review articles, case reports, or case series of less than five patients, (c) studies with abstracts only and (d) animal/cell studies.

### 2.3. Search Approach

We searched databases comprehensively (PubMed, Scopus, Cochrane Library, Web of Science and Clinicaltrial.gov) from the inception of targeted therapy until December 2024. The search keywords were: (“anaplastic thyroid cancer” [MeSH Terms] OR (“anaplastic” [All Fields] AND “thyroid cancer” [All Fields]) OR “thyroid cancer” [All Fields] AND (“tyrosine kinase inhibitors” [“Lenvatinib, Sorafenib, Pazopanib +/− RT, Gefitinib, Apatinib, Sunitinib, Imatinib”]) [All fields], (“Fosbretabulin”) [All fields], AND (“BRAF/MEK inhibitors” [“Vemurafenib, Dabrafenib/Trametinib, Encorafenib/binimetinib”]) [All fields], AND (“Immunotherapy” [“nivolumab, pembrolizumab, nivolumab–ipilimumab, atezolizumab, Spartalizumab”]) +/− “Lenvatinib” +/− “radiotherapy” [All fields], AND (“mTOR inhibitor” [“Everolimus, Sapanisertib”]) [All fields], AND (“TRK inhibitors” [“Larotrectinib, Entrectinib”]) [All fields], AND (“RET inhibitor” [“Selpercatinib”]) [All fields], and (“chemotherapy with Paclitaxel with Carboplatin or radiotherapy”) [All fields]. We also manually searched for any potential eligible study.

We used Endnote (Clarivate Analytics, Philadelphia, PA, USA) to remove duplicate data, and all retrieved studies were screened for titles/abstracts and full-text articles to assess relevance to the present meta-analysis. The publications reporting the highest number of patients were included in the studies derived from the same centre, with the possibility of data overlap.

### 2.4. Data Extraction

Two investigators independently reviewed the study selection. The investigators resolved any discrepancies through discussion. We recorded the information on the following attributes of included studies: previously given treatments, criteria for tumour response [complete response (CR), partial response (PR), stable disease (SD), overall response rate (ORR), disease control rate (DCR)], toxicity profile, biomarker analysis and reported progression-free survival (PFS) and overall survival (OS) rates.

### 2.5. Quality Assessment

A methodological index for non-randomised studies (MINORS) and a JBI critical appraisal checklist were applied to single-arm and retrospective studies [14,15]. The MINORS score consists of 8 items that are zero if not reported, 1 point when reported but inadequate and 2 points when adequately said. The maximum MINOR score is 16 points, and we categorised as poor a score <8, fair = 8 to 12 while excellent 13 to 16 scored points.

### 2.6. Statistical Methods

Pooled PR, SD and ORR analysis was performed using R version 3.6.3. For binary endpoints (PFS and OS), we used arcsine log transformation along with inverse probability weighting (IPW). The random-effect model was utilised because it harboured a more significant standard error in the pooled estimate, making it suitable for conflicting or erratic estimates. The chi-square test (Cochrane Q test) calculated I-squared (*I*^2^) to measure heterogeneity among studies. A *p* value less than 0.1 was considered significant heterogeneity. *I*^2^ values ≥ 50% were suggestive of high heterogeneity. Publication bias was calculated using Begg’s test and Egger’s test. A *p* value < 0.05 was weighed up as statistically significant.

## 3. Results

The electronic search revealed 619 relevant citations. After screening, 263 full-text articles were retrieved for further assessment. Finally, 47 studies (26 prospective phase II trials met MINORS and 21 retrospective studies met JBI critical appraisal checklist criteria) were identified (Figure 1) (Appendix A, Table A1 and Table A2); the total population of these studies was 980 patients [8,9,16,17,18,19,20,21,22,23,24,25,26,27,28,29,30,31,32,33,34,35,36,37,38,39,40,41,42,43,44,45,46,47,48,49,50,51,52,53,54,55,56,57,58,59,60].

The median age of the whole population was 62.9 years (range: 58–77). The majority of the population was previously heavily treated. Patient characteristics are given in Table 1 (Appendix A Table A3).

All studies examined were published from 2008 to 2024. Lenvatinib monotherapy was used in 11/47 studies (23.4%) and with pembrolizumab in five studies (10.6%). BRAF inhibitors, with DT, were used in ten studies (21.3%), Encorafenib plus binimetinib in one (2.1%) and vemurafenib in one study (2.1%). IO, +/− TKI, or DT radiation, was used in ten studies (21.3%).

### 3.1. Overall Response Rates (ORR) and Disease Control Rates (DCR)

The pooled ORR was seen in 291/980 patients (29.7% [95% CI, 25.4–34.2%; *I*^2^ = 42.4%; *p* = 0.0009]) (Figure 2).

In a subgroup analysis, studies using DT showed a significantly higher pooled ORR of 64.9% (95% CI, 46.3–83.4%; *I*^2^ = 77.5%; *p* < 0.001), while in studies using lenvatinib plus pemrolizumab (LP), the pooled ORR was 42% (95% CI, 28.6–55.3%; *I*^2^ = 55.8%; *p* = 0.04). Lenvatinib and other TKI monotherapy or IO alone did not result in meaningful ORR.

The pooled DCR was seen in 497 patients (51.0% [95% CI, 44.0–59%; *I*^2^ = 85.2%; *p* < 0.0001]). SD was obtained in 206 (21.0%). In the subgroup analysis, higher DCR rates were seen with DT (74.4% [95% CI, 57.0–81.0%]) and LP (64.2% [95% CI, 48.0–78.0%]).

### 3.2. Survival Outcomes

In total, 489/980 (49.9%) deaths were reported. A significant number of patients remained alive relative to baseline (mean difference [MD], 2.07 [95% CI: 1.90–2.24; *I*^2^ = 88.6%; *p* < 0.0001]) (Figure 3).

The pooled median OS was 7.2 months (95% CI: 5.6–8.8 months *I*^2^ = 87.8%; *p* < 0.0001); the cumulative OS rate at 6 months, 9 months, 12 months and 24 months was 58.2%, 44.3%, 38.5% and 21.2%, respectively (Appendix A Figure A1). In the subgroup analysis, improved OS rates were observed in DT with and without pembrolizumab (median OS 11.2 months, 95% CI: 5.3–17.1 months) and LP regimens (median OS 14.4 months, 95% CI: 7.6–21.1 months). Only two studies of lenvatinib monotherapy showed any OS benefit [16,17].

A total of 329/980 (33.6%) PFS outcomes were reported. Pooled PFS significantly improved relative to baseline (MD 1.50 [95% CI: 1.33–1.68, *I*^2^ = 90.9%; *p* < 0.0001]), while the median PFS was 5.4 months (95% CI, 4.0–6.7 months); *I*^2^ = 97.9%; *p* < 0.0001 (Figure 4), with the cumulative PFS rate at 6, 9, 12 and 24 months being 47.2%, 32.7%, 26.1% and 9.2%, respectively.

In the subgroup analysis, improved PFS rates were seen with DT with and without pembrolizumab (median PFS 12.1 months, 95% CI: 7.5–15.9 months) and LP (median PFS 10.0 months, 95% CI: 5.3–14.6 months). Only one study of lenvatinib monotherapy resulted in PFS improvement [16].

RT was administered to 76 out of 980 patients (7.2%). The RT regimen included a daily dose of 2 Gy per fraction over 33 days (totalling 66 Gy) combined with weekly paclitaxel, with or without pazopanib, in 71 patients ORR 31%) [30]. Alternatively, a bi-daily fractionation schedule was used in 5 patients [53], delivering 3.5 Gy per fraction at intervals of more than 6 h over 2 consecutive days (totalling 14 Gy) combined with pembrolizumab (ORR 40%).

### 3.3. Biomarker Analysis

Biomarker analysis was performed in 24/47 studies (51.1%) (Appendix A Table A3).

BRAF V600 mutations were reported in 18 out of 47 studies (38.3%), and 144 out of 980 patients (14.7%) were identified as BRAF mutated.

-PIK3CA status was mentioned in 4 studies (8.5%) and was positive in 10 patients (1.0%).-RAS (KRAS and NRAS) status was reported in 4 studies (8.5%) and was positive in 11 patients (1.1%).-NTRK1/3 was mentioned in 1 study (2.1%) and was positive in 7 patients (0.7%).-P53 status was reported in 3 studies (6.3%) and was detected in 10 patients (1.0%).-Programmed death ligand-1 (PDL-1) status was reported in 10 studies (21.3%) and was positive in 90 patients (9.2%).

### 3.4. Toxicity Profile

Grade 3 and above toxicity was reported for 928/980 (94.7%) patients (Table 2).

In lenvatinib studies, the predominant grade 3 and above toxicities were hypertension (16.4%), diarrhoea (11.3%), loss of appetite (9.7%) and hand-foot syndrome (8.7%). Dose modifications, interruptions and withdrawal of lenvatinib were observed at rates of 30.2%, 22.6% and 8.7%, respectively.

For DT studies, notable grade 3 and above toxicities included anaemia (11.1%), fatigue (8.7%), hypertension (6.7%) and pneumonitis (6.77%). Dose adjustments, interruptions and withdrawal of DT occurred at 15%, 16.5% and 4.4%, respectively. Immunotherapy studies revealed lipase/amylase elevation (9.1%), adrenal insufficiency (3.0%), dermatitis (2.1%) and acute heart failure (2.1%) as common grade 3 and above toxicities.

The resultant funnel plot showed significant heterogeneity (Egger’s test *p* < 0.001) (Appendix A Figure A2).

## 4. Discussion

Locally advanced, recurrent and metastatic ATC remains one of the most challenging malignancies due to its highly aggressive nature and historically limited therapeutic options, which have primarily included systemic chemotherapy and neck irradiation. More recently, there has been a shift towards targeted therapies and IO, offering new avenues for treatment.

This meta-analysis represents the first comprehensive evaluation of various therapeutic approaches in ATC, encompassing mutationally targeted therapies, IO and novel chemotherapeutic agents to provide insights into therapeutic decision-making within this complex treatment landscape. In contrast, the prior meta-analysis by Huang D et al. focused solely on the efficacy of lenvatinib [7].

The observed ORR of 64.9% in patients receiving dual-targeted therapy (DT) underscores the potent efficacy of BRAF/MEK inhibition, consistent with preclinical and clinical data demonstrating the oncogenic reliance of ATC on the MAPK pathway [36,37]. Similarly, the combination of lenvatinib and pembrolizumab achieved an ORR of 42%, supporting the therapeutic potential of combining angiogenesis inhibition with immune checkpoint blockade in ATC. DCRs were also notable, with DT achieving 74.4% and lenvatinib plus pembrolizumab reaching 64.2%, suggesting that durable disease stabilisation is possible in patients harbouring actionable mutations. However, despite these encouraging response rates, the OS remains limited at 7.2 months, with the most extended median OS survival observed in DT-treated patients (11.2 months) and those receiving lenvatinib atinib plus pembrolizumab (14.4 months).

### 4.1. Next-Generation Sequencing (NGS) and Molecular-Driven Treatment

Current guidelines emphasise the routine use of biomarker analysis [61,62], given the high prevalence of genomic alterations in ATC. The most frequently observed mutations include BRAF V600E and MEK alterations (40.50%) [38], TP53 (63%) [55], RET and RAS mutations (22%) [56] and TERT promoter mutations (75%) [62]. Additionally, alterations in PIK3CA (18%) [53], EIF1AX (14%) and PTEN (14%) [63] have been documented, while the prevalence of NTRK fusions and other “non-actionable mutations” such as “NESTIN, CCND1, POU5F1, MCL1, MYBL2, IQGAP1, SOX2 and NANOGâ€” remains unknown [64].

In the present meta-analysis, biomarker assessment was reported in only 51.1% of the included studies. Among the patient population, 14.7% were identified as BRAF V600E-mutated, while 9.2% were PD-L1 positive. NTRK1/3 fusions were detected in only 0.7% of cases (12 ATC cases among 83 total NTRK fusion-positive tumours), highlighting a rare but clinically significant subgroup that may benefit from TRK inhibitors. In this meta-analysis, entrectinib demonstrated an ORR of 20%, while larotrectinib achieved an ORR of 29%.

Additionally, the presence of RET fusions in some studies supports the potential role of selective RET inhibitors such as selpercatinib, as evidenced in the LIBRETTO-001 trial [65]. However, this trial was not included in the current meta-analysis, as it only enrolled two ATC patients among 166 individuals with RET-driven tumours. The detection of PIK3CA (1.0%) and RAS mutations (1.1%) suggests alternative pathways for targeted therapy. However, their clinical relevance remains uncertain due to the absence of approved targeted agents for these mutations in ATC. These findings highlight the necessity of incorporating comprehensive genomic profiling into routine clinical practice to optimise treatment selection and improve patient outcomes. The limited availability of molecular data in many studies underscores the need for standardised biomarker assessment in ATC to define treatment strategies better and identify patients most likely to benefit from mutational driven and IO approaches.

### 4.2. PD-L1 Expression in ATC

A total 21.3% of included studies reported PD-L1 expression, with 9.2% of ATC patients identified as PD-L1 positive. This is pretty low, as 22–29% of ATC tumour samples have been reported to express PD-L1 [66].

One of the primary challenges in utilising PD-L1 as a predictive biomarker in ATC is the absence of standardised cutoff values for clinical decision-making. Unlike non-small-cell lung cancer (NSCLC) or head and neck squamous cell carcinoma (HNSCC), where PD-L1 expression thresholds (1–50%) have been validated for pembrolizumab, no established ATC-specific cutoff values currently exist [67].

While our analysis demonstrates that lenvatinib combined with pembrolizumab achieves an ORR of 42% and a median OS of 14.4 months, these findings emphasise further investigation. Future clinical trials should prioritise standardising PD-L1 scoring thresholds and explore potential synergistic treatment strategies, including its combination with angiogenesis inhibitors or RT, to enhance IO efficacy in ATC.

### 4.3. Radiation Therapy in ATC

In this meta-analysis, RT was administered only to 7.2% of patients using two fractionation regimens. Most patients (n = 71) received conventional fractionation, consisting of a total dose of 66 Gy delivered in 33 fractions (2 Gy per fraction) over 33 days, combined with weekly paclitaxel, with or without pazopanib. In contrast, a smaller cohort (n = 5) underwent hypofractionated RT (HypoRT), receiving 14 Gy in 4 fractions (3.5 Gy per fraction) over two consecutive days in combination with pembrolizumab. HypoRT achieved a higher ORR (40%) than conventional RT (31%) [30,53].

This finding aligns with recent work by Oliinyk et al., [68] who evaluated the use of hypoRT in ATC patients treated with 3DRT or IMRT. In their cohort of 17 ATC patients, a cumulative radiation dose of <30 Gy was delivered, with four patients receiving concurrent chemotherapy (carboplatin with paclitaxel or doxorubicin weekly). The median OS was four months (range: 1–12 months), with survival rates of 82%, 55% and 36% at one, three and six months, respectively. Subsequent authors performed a systematic review of hypoRT in ATC, supporting its role as an integral component of multimodal treatment and demonstrating promising clinical outcomes. Given the increasing evidence favouring hypoRT, its incorporation into future clinical trials should be explored, particularly in combination with molecularly targeted agents and IO.

Toxicity profiles, which differed across various therapeutic approaches, were also examined. The studies involving Lenvatinib demonstrated the highest incidence of grade 3 and above adverse effects among all studies, like hypertension (16.4%) and diarrhoea (11.3%), leading to a higher frequency of dose modifications and interruptions. DT studies exhibited anaemia (11.1%) as a significant adverse effect. IO studies revealed lipase/amylase elevation (9.4%).

A recent meta-analysis by Cao and colleagues [69], based on six prospective trials of TKI monotherapy involving a pooled population of 140 ATC patients, reported a median OS of 4.8 months and a median PFS of 2.6 months. In contrast, our meta-analysis demonstrated a median OS of 7.2 months and a median PFS of 5.4 months, highlighting the enhanced efficacy of dual therapy (DT) with or without immunotherapy (IO) and combinations of TKI with IO. Combination therapy as a potential therapeutic direction for ATC patients is also supported by the recently published study by Zheng X et al. using TKI (erlotinib) with chemotherapy [70]. This study was not included in meta-analysis due to lack of full details.

The strengths of the present meta-analysis were as follows: (1) a comprehensive literature search, (2) rigorous inclusion and exclusion criteria, (3) a large sample size with inclusion of 47 studies involving 980 patients contributes to the robustness of the analysis, providing a substantial dataset.

Limitations: Although IPW and standard effect size were used to reduce bias and enable comparisons, the limited number of included randomised studies and variability in study designs among the included studies may have influenced the OS and PFS outcomes.

## 5. Conclusions

This meta-analysis comprehensively evaluated therapeutic strategies for locally advanced, recurrent and metastatic ATC, focusing on novel mutational-targeted therapy, IO, chemotherapy and RT. While dual-targeted therapy (BRAF/MEK inhibition) demonstrated the highest ORR at 64.9% and lenvatinib plus pembrolizumab achieved a 42% ORR, OS remains a challenge, with median OS reaching only 11.2 months for DT and 14.4 months for lenvatinib plus pembrolizumab. Despite these improvements in treatment outcomes, our meta-analysis showed underutilisation of biomarker-driven treatment, with genomic profiling reported in just over half of the included studies (51.1%), highlighting the critical need for standardised molecular testing. The limited data regarding PD-L1 expression (9.2% of patients) and the absence of validated cutoffs in ATC further complicate patient selection for IO. HypoRT showed promise with a higher ORR (40%) than conventional RT (31%), suggesting its potential within multimodal approaches. While combination therapy appeared superior compared to historical TKI monotherapy data, toxicity profiles varied across treatments, necessitating careful monitoring. These findings underscore the urgent need for well-designed prospective trials investigating optimal sequencing and combinations of DT, IO and RT, incorporating standardised molecular profiling (including NGS) to guide treatment decisions, validating PD-L1 cutoffs for IO and further evaluating the role of HypoRT in combination with systemic therapies. Such trials should also prioritise evaluating toxicity management strategies and exploring novel mutational-driven and targeted therapies to improve long-term outcomes for ATC patients.

## Figures and Tables

**Figure 1 cancers-17-00777-f001:**
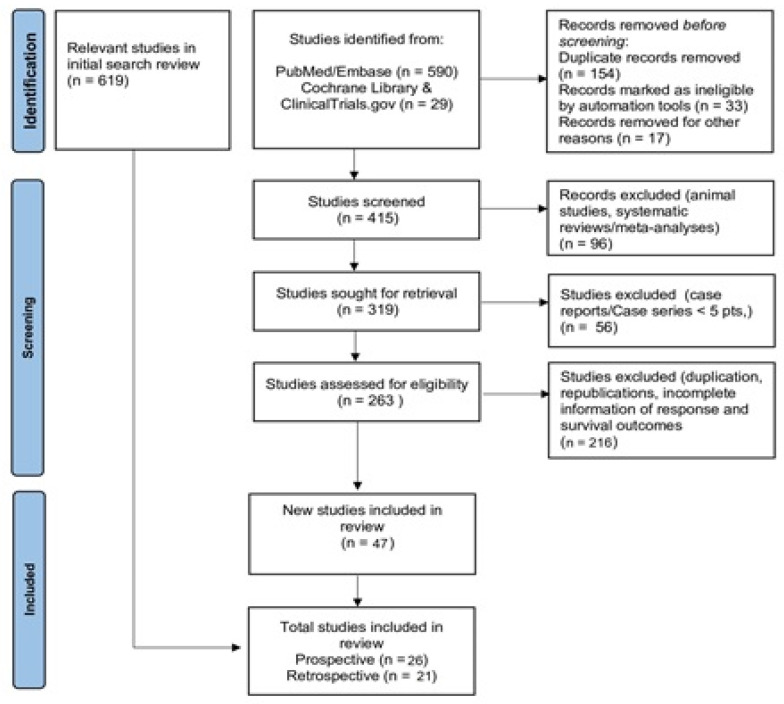
Flow diagram of the studies’ selection strategy.

**Figure 2 cancers-17-00777-f002:**
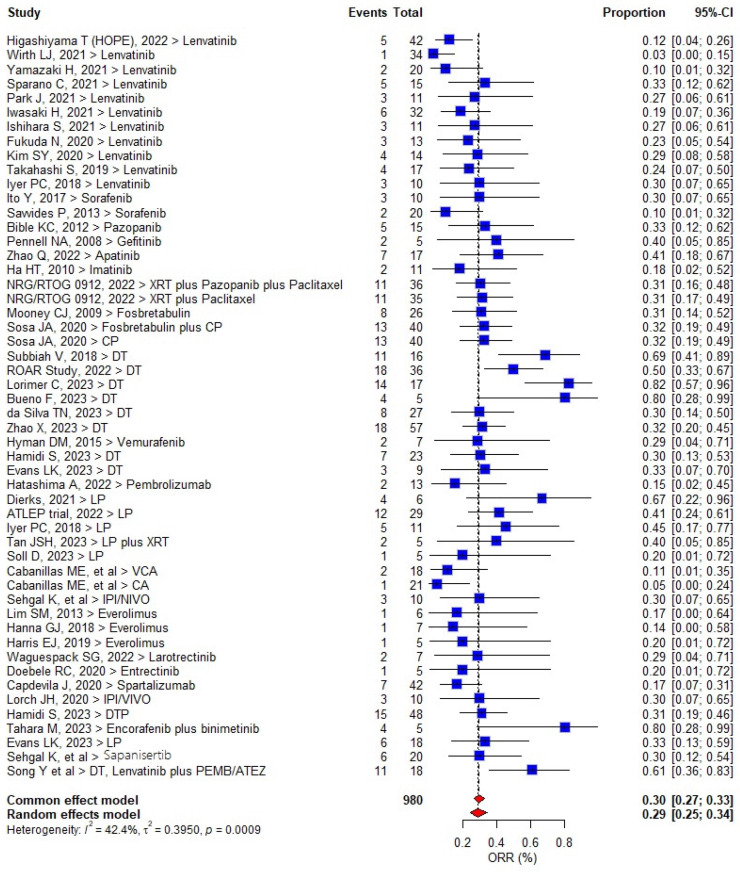
Forest plot of overall response rates in included studies.

**Figure 3 cancers-17-00777-f003:**
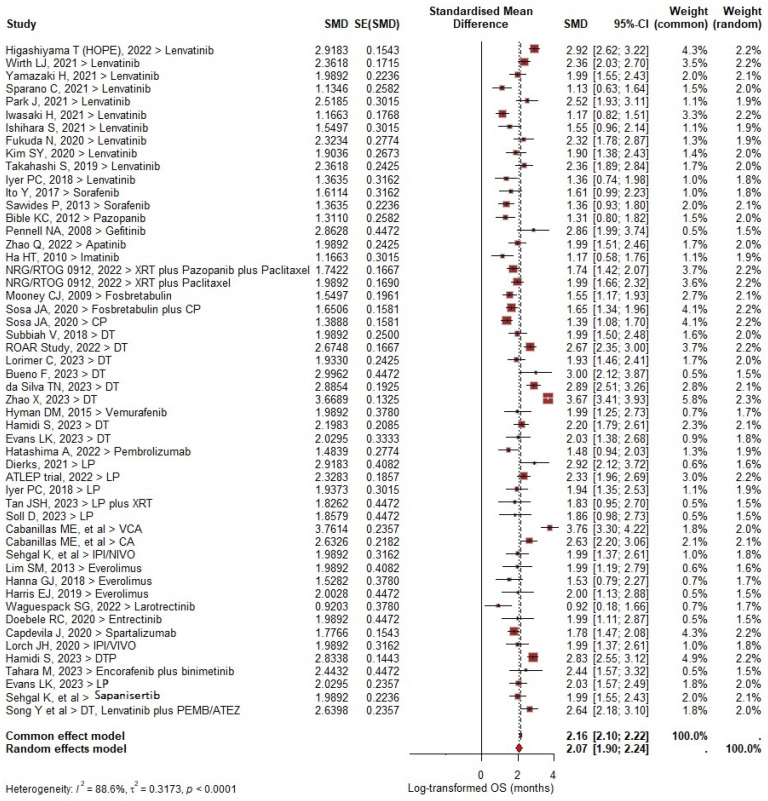
Forest plot of overall survival in included studies.

**Figure 4 cancers-17-00777-f004:**
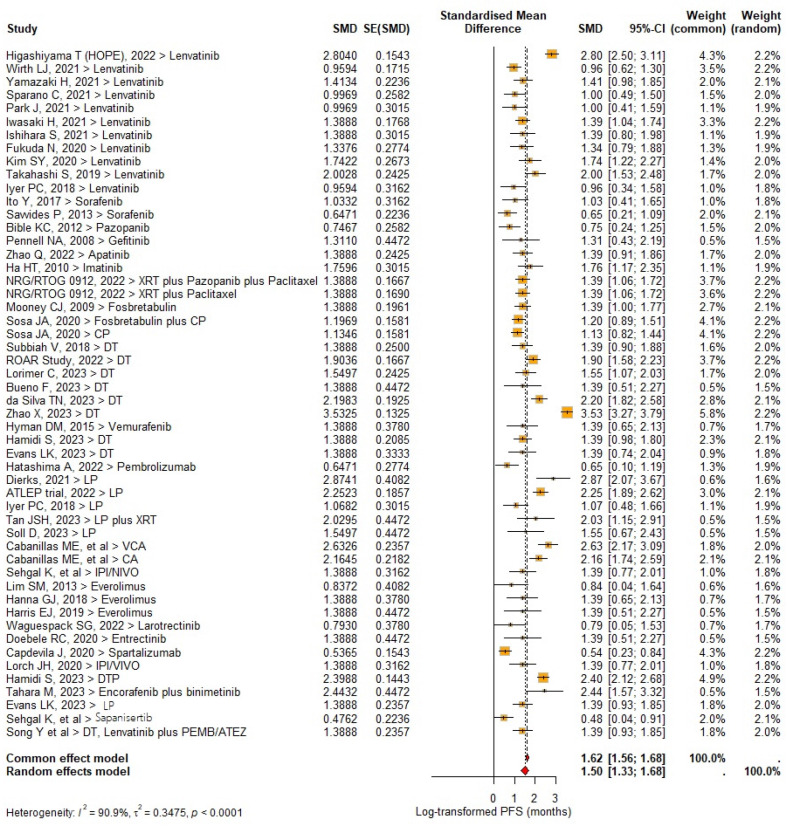
Forest plot of progression survival rate in included studies.

**Table 1 cancers-17-00777-t001:** Characteristics of included studies.

Study/Year/ Nature	Country	N (Median Age)	TNM (n)	Previous Treatment (%)	Intervention	Endpoints	ORR (%)	Median PFS (Months)	Median OS (Months)
**Higashiyama T, et al. (HOPE) [16]** **Prospective** **2022**	Japan	42 (73y)	T4 = 37 N1a = 3 N1b = 23 M0 = 17 M1 = 21	CTH = 40 XRT = 21.4 Sx = 54.7	Lenvatinib	Primary: OS Secondary: PFS, ORR, DCR, safety	5 (11.9)	16.5	18.5
**Wirth LJ, et al. [17]** **Prospective** **2021**	US, UK, France, Italy, Australia,	34 (65y)	M1 = 34	CTH = 62 TKI = 9 IO = 9 XRT = 65 Sx = 71	Lenvatinib	Primary: ORR, safety Secondary: PFS, OS	1 (2.9)	2.6 (1.4–2.8)	10.6 (3.8–19.8)
**Yamazaki H, et al. [18]** **Retrospective** **2021**	Japan	20 (73.6y)	T4 = 14	Sx = 6	Lenvatinib	-	2 (10.0)	4.1 (1.1–12.2)	-
**Sparano C, et al. [19]** **Retrospective** **2021**	France, Italy	15 (63y)	Recurrent/ M1 = 15	XRT = 71 CTH = 73.2 TKI = 26.5	Lenvatinib	-	5 (33.3)	2.7 (1.9–3.5)	3.1 (0.6–5.5)
**Park J, et al. [20]** **Retrospective** **2021**	South Korea	11 (66.4y)	T4/ M1 = 11	Sx = 64.2 XRT = 53.3 CTH = 24.2 TKI = 15.8	Lenvatinib	-	3 (27.3)	2.7 (0.1–12.7)	12.4 (1.7–47.7)
**Iwasaki H, et al. [21]** **Retrospective** **2021**	Japan	32 (77y)	T4/ M1 = 32	-	Lenvatinib	-	6 (18.8)	-	3.2 (0.5–28.9)
**Ishihara S, et al. [22]** **Retrospective** **2021**	Japan	11 (74y)	Recurrent = 5 T4 = 4 T3b = 2	CTH = 63.6 XRT = 63.6 TKI = 27.3	Lenvatinib	-	3 (30.0)	-	4.7 (1.9–13.1)
**Fukuda, N et al. [23]** **Retrospective** **2020**	Japan	13 (68y)	T4 = 10 N1 = 7 M1 = 11	Sx = 77 XRT = 46 CTH = 31	Lenvatinib	-	3 (23.0)	3.8 (1.8–6.4)	10.2 (3.7–17.6)
**Kim M, et al. [24]** **Retrospective** **2020**	South Korea	14 (64.9y)	T4 = 14	Sx = 33 CTH = 100 XRT = 100	Lenvatinib	-	4 (28.6)	5.7 (2.2–8.3)	6.7 (3.0–8.4)
**Takahashi S, et al. [25]** **Prospective** **2019**	Japan, US	17 (65y)	T4 = 17	Sx = 82 CTH = 41 XRT = 53	Lenvatinib	Primary: Safety Secondary: PFS, ORR, DCR, OS	4 (23.5)	7.4 (1.7–12.9)	10.6 (3.8–19.8)
**Iyer PC, et al. [26]** **Retrospective** **2018**	US	10 (67y)	T3-4/N1 = 4 M1 = 6	Sx = 50 CTH = 60 XRT = 44	Lenvatinib	-	3 (30.0)	2.6 (2.8-NR)	3.9 (2.5-NR)
**Ito Y et al. [27]** **Prospective** **2017**	Japan	10 (72y)	M1 = 8 T4 = 2	Sx = 70 CTH = 60 XRT = 70	Sorafenib	Primary: Safety Secondary: PFS, ORR, DCR, OS	3 (40.0)	2.8 (0.7–5.6)	5 (0.7–5.7)
**Savvides P, et al. [28]** **Prospective** **2013**	US	20 (59y)	M1 = 20	CTH = 100 XRT = 90 Sx = 90	Sorafenib	Primary: ORR, DCR Secondary: PFS, OS, safety	2 (10.0)	1.9 (1.3–3.6)	3.9 (2.2–7.1)
**Bible KC, et al. [29]** **Prospective** **2012**	US	15 (63)	T4 = 2 N1 = 1 M1 = 12	CTH = 100 XRT = 90 Sx = 90	Pazopanib	Primary: ORR, DCR Secondary: PFS, OS, safety	5 (33.3)	2.1 (NR)	3.7 (0.5–35)
**NRG/RTOG 0912** [30] **Prospective 2023**	US	71 (63y)	T4 = 71 N1 = 52 M1 = 26	Sx = 57.5	a. Pazopanib + weekly Paclitaxel +IMRT 66 Gy/33 fractions (36pts) b. Weekly Paclitaxel +IMRT 66 Gy/33 fractions	Primary: OS Secondary: PFS, safety	11 (30.5) 11 (31.4)	-	5.7 (4.0–12.8) 7·3 (4.3–10.6)
**Pennell NA, et al.** [31] **Prospective 2008**	US	5/27 (65y)	T4/M1 = 5	CTH = 22 Sx = 94 XRT = 85	Gefitinib	Primary: ORR, DCR Secondary: PFS, OS, safety	2 (40.0)	3.7	17.5
**Zhao Q, et al.** [32] **Prospective 2022**	China	17 (61y)	T4 = 3 N1 = 7 M1 = 6	-	Apatinib	Primary: ORR, DCR	7 (41.1)	-	-
**Ha HT, et al. [33]** **Prospective** **2010**	US	11 (65y)	Recurrent = 7 M1 = 7	Sx = 64 CTH = 55	Imatinib	Primary: ORR	2 (18.2)	5.8 (2–29)	3.2 (2–37)
**Mooney CJ, et al. [34]** **Prospective** **2009**	US	26 (59y)	Recurrent = 24 M1 = 7	CTH = 50 Sx = 61 XRT = 61	Fosbretabulin	Primary: Safety Secondary:	8 (30.7)	-	4.7 (2.5–6.4)
**Sosa JA, et al. [35]** **Prospective** **2014**	US	80 (63y)	Recurrent = 20	Sx = 25 XRT = 13.8 CTH =5	Fosbretabulin +CP Or CP alone	Primary: OS	13 (32.5) 13 (32.5)	3.3 (2.3–5.6) 3.1 (2.7–5.4)	5.2 (3.1–9) 4.0 (2.8–6.2)
**Subbiah V, et al. [36]** **Prospective** **2018**	US	16 (72y)	Recurrent = 16	Sx = 88 XRT = 81 CTH = 38	DT	Primary: ORR, DCR Secondary: PFS, OS, safety	11 (68.7)	NA	NA
**ROAR Study [37]** **Prospective** **2022**	US, France, south Korea	36 (71y)	T4 = 1 M1 = 35	Sx = 83 XRT = 83 CTH = 42 RAI = 31 TKI = 19 IO = 11	DT	Primary: ORR Secondary: PFS, OS, DOR	18 (50)	6.7 (4.7–13.8)	14.5 (6.8–23.2)
**Lorimer C, et al. [8]** **Retrospective** **2023**	UK	17 (68y)	T4/M1 = 17	Sx = 59 CTH = 12 RAI = 12 XRT = 18	DT	-	14 (82.2)	4.7 (1.4–7.8)	6.9 (2.46-NR)
**Bueno F, et al. [38]** **Retrospective** **2023**	Argentina	5 (70y)	M1 = 5	Sx = 60 XRT = 20	DT	-	4 (80.0)	-	20 (18-NR)
**da Silva TN, et al. [39]** **retrospective** **2023**	Portugal	27 (77y)	Recurrent/ M1 = 27	Sx = 40.7 XRT = 40.7 CTH = 25.9 TKI = 7.4	DT	-	8 (29.6)	9.0 (4.9–13.0)	17.9 (15.9–19.8)
**Zhao X, et al. [40]** **Retrospective** **2023**	US	57 (67.2y)	T4 = 20 M1 = 37	IO = 75.4 XRT = 47.4	DT	-	18 (31.5)	34.2 (15.8–NA)	39.2 (NR)
**Hyman DM, et al. [41]** **Prospective** **2015**	US, UK, Germany, France, Spain	7/122 (65y)	Recurrent/ M1 = 7	Any = 78 XRT = 67	Vemurafenib	Primary: ORR Secondary: PFS, OS	2 (29.0)	-	-
**Hatashima A, et al. [42]** **Retrospective** **2022**	US	13 (70y)	T4 = 2 M1 = 11	-	Pembrolizumab (12 patients) Nivolumab (1 patient)	-	2 (16.0)	1.9	4.4 (4.0–29)
**Capdevila J, et al. [43]** **Prospective** **2020**	US, Canada, Germany, Italy, Switzerland, France, Poland	42 (62.5y)	Recurrent/M1 = 42	XRT = 71.4 Sx = 66.7	Spartalizumab	Primary: ORR Secondary: Safety	7 (16.6)	1.7 (1.2–1.9)	5.9 (2.4-NR)
**Lorch JH, et al. [44]** **Prospective** **2020**	US	10/49 (65y)	T4/M1 = 10	-	Ipilimumab + Nivolumab	Primary: ORR	3 (30.0)	-	-
**Dierks C, et al. [45]** **Retrospective 2021**	Germany	6 (63.5y)	Recurrent/ M1 = 6	Sx = 100 XRT = 87.5 CTH = 75 RAI = 25	LP	-	4 (66.0)	17.7	18.5
**ATLEP trial [46]** **Prospective** **2022**	Germany	29 (63y)	N1/M1 = 29	Sx = 90 XRT = 90 CTH = 90	LP		12 (41.2)	9.5	10.25 (NR)
**Iyer PC, et al. [47]** **Retrospective** **2018**	US	11 (67y)	T4/M1 = 11	Sx = 50 XRT = 44 CTH = 69	LP	-	5 (42.0)	2.9 (2.2–3.7)	6.93 (1.7–12.1)
**Lim SM, et al. [48]** **Prospective** **2013**	South Korea	6/40 (61y)	T4/M1 = 6	Sx = 68 CTH = 22 XRT = 15 TKI = 2	Everolimus	Primary: ORR Secondary: PFS, OS	1 (16.0)	2.3 (1.1–3.7)	-
**Hanna GJ, et al. [49]** **Prospective** **2018**	US	7/50 (62y)	Recurrent/ M1 = 7	Sx = 71 XRT = 57 RAI = 28 CTH = 28 TKI = 14	Everolimus	Primary: ORR Secondary: Safety	1 (14.2)	-	4.6 (<1–29.9)
**Harris EJ, et al. [50]** **Retrospective** **2019**	US	5 (75y)	Recurrent/ M1 = 5	XRT = 60 CTH = 60 RAI = 40	Everolimus	-	1 (20.0)	-	7.4 (<1–40)
**Waguespack SG, et al. [9]** **Prospective** **2022**	US, Canada, Ireland, Italy, Germany, S. Korea	7/29 (60y)	Recurrent/ T4 = 7	Sx =100 XRT = 71 CTH = 43	Larotrectinib	Primary: ORR Secondary: PFS, OS, Safety	2 (29.0)	2.2 (<1–6)	2.5 (<1–6)
**Doebele RC, et al. [51]** **Prospective** **2020**	US, UK, Australia, Italy, Hong Kong, Spain, S. Korea, Poland, Netherlands	5/54 (58y)	-	-	Entrectinib	Primary: ORR	1 (20.0)	-	-
**Hamidi S, et al.** [52] **Retrospective** **2024**	US	71	T4 = 23 M1 = 48	Sx/XRT	23 = DT 48 = DTP	Primary: OS Secondary: PFS	7 (30.4) 15 (31.3)	DT = 4 DTP =11	DT = 9 DTP = 17
**Tan JSH, et al.** [53] **Retrospective** **2024**	Singapore	5 (63.6y)	T4 = 3 M1 = 2	-	Quad-XRT/LP = 2. Quad-XRT/P = 3	Primary: OS Secondary: PFS, ORR	2 (40.0)	7.6	6.2
**Tahara M, et al.** [54] **Prospective** **2024**	Japan	5/22	T4/M1 = 5	-	Encorafenib plus binimetinib	Primary: OS Secondary: PFS, ORR	4 (80.0)	11.5	11.5
**Evans LK, et al.** [55] **Retrospective** **2025**	US	41 (67.4y)	Recurrent/T4/M1 = 41	-	LP = 18 DT = 9	Primary: OS	6 (33.3) 3 (33.3)	-	7.6
**Soll D, et al.** [56] **Retrospective** **2024**	Germany	5 (65y)	M1 = 5	-	LP	Primary: OS Secondary: PFS	1 (20.0)	4.7	6.4
**Cabanillas ME, et al.** [57] **Prospective** **2024**	US	39	T4 = 12 M1 = 30	XRT = 12 CTH = 38.1	VCA = 18 CA = 21	Primary: OS Secondary: PFS	2 (11.1) 1 (5.7)	VCA = 13.9 CA = 8.7 BA = 6.2	VCA = 43 CA = 13.9 BA = 8.7
**Sehgal K, et al.** [58] **Prospective** **2024**	US	20/26	Recurrent /M1 = 20	TKI = 20	Sapanisertib	Primary: ORR, PFS	6 (30.0)	1.6	-
**Song Y, et al.** [59] **Retrospective** **2024**	China	18	T4/M1	-	DTP, LP, anlotinib sintilimab, or camrelizumab	Primary: ORR, PFS, OS	11 (61.1)	-	14
**Sehgal K, et al.** [60] **Prospective** **2024**	US	10/49 (65y)	Recurrent/metastatic/T4 = 10	XRT =80 TKI = 70	Ipilimumab/ nivolumab	Primary: ORR Secondary: PFS, OS, Safety	3 (30.0)	-	-

N = number of patients, TNM = tumour, node, stage = CR = complete response, PR = partial response, ORR, overall response rate, DCR = disease control rate, PFS = progression-free survival, OS = overall survival, CTH = chemotherapy, XRT = radiation therapy, Sx = surgery, RAI = radioactive iodine, TKI = tyrosine kinase inhibitors, IMRT = intensity modulated radiation therapy. CP = carboplatin/paclitaxel, D/D = docetaxel/doxorubicin, DT = dabrafenib/trametinib, DTP = dabrafenib/trametinib/pembrolizumab, VCA = vemurafenib/cobimetinib plus atezolizumab, CA = cobimetinib plus atezolizumab.

**Table 2 cancers-17-00777-t002:** Toxicity Profile of Included Studies.

G3/>Toxicity	Lenvatinib (n = 266)	Sorafenib (n = 30)	Pazopanib (n = 51)	Gefitinib (n = 5)	Apatinib (n = 17)	Imatinib (n = 11)	Chemotherapy (n = 81)	mTORi (n = 18)	BRAFi (n = 206)	IO (n = 232)	NTRKi (n = 11)
Loss of appetite	29 (11.0%)	1 (3.0%)	2 (5.8%)	2 (40.0%)	-	1 (9.0%)	2 (2.3%)	5 (27.7%)	10 (4.8%)	4 (1.7%)	1 (9.0%)
Weight loss	23 (8.6%)	-	-	1 (20.0%)	-	1 (9.0%)	1 (1.2%)	4 (22.2%)	8 (3.8%)	4 (1.7%)	1 (9.0%)
Fatigue	21 (7.9%)	-	7 (13.7%)	1 (20.0%)	-	-	1 (1.2%)	1 (5.5%)	18 (8.7%)	4 (1.7%)	3 (27.3%)
Hypertension	37 (14.0%)	-	1 (1.9%)	-	6 (35.3%)	-	2 (2.3%)	-	14 (6.7%)	-	-
HFS	20 (7.5%)	3 (9.0%)	-	-	-	-	-	-		4 (1.7%)	-
Nausea	12 (4.5%)	2 (6.0%)	4 (7.8%)	1 (20.0%)	-	-	11 (13.6%)	5 (27.7%)	8 (3.8%)	-	1 (9.0%)
Diarrhoea	24 (9.0%)	3 (9.0%)	-	3 (60.0%)	-	-	12 (14.8%)	-	8 (3.8%)	-	1 (9.0%)
Anaemia	3 (1.1%)	-	2 (3.9%)	-	-	2 (18.0%)	8 (9.8%)	-	23 (11.1%)	2 (0.8%)	1 (9.0%)
Thrombocytopenia	8 (3.0%)	-	1 (1.9%)	-	1 (5.9%)	5 (45.5%)	14 (17.3%)	-	-	-	-
Lymphopenia	4 (1.5%)	-	5 (9.8%)	-	1 (5.9%)	-	8 (9.8%)	2 (11.0%)	-	-	-
Neutropenia	2 (1.0%)	-	7 (13.7%)	-	1 (5.9%)	1 (9.0%)	2 (2.3%)	5 (27.7%)	1 (0.5%)	-	-
Hypophosphatemia	-	1 (3.0%)	1 (1.9%)	-	-	-		2 (11.0%)	-	2 (0.8%)	-
Hyperglycaemia	-		-	-	-	-	1 (1.2%)	2 (11.0%)	8 (3.8%)	2 (0.8%)	-
Hyponatremia	-	2 (6.0%)	-	-	-	-	-	-	11 (5.3%)	3 (1.3%)	-
Voice alteration	3 (1.1%)		3 (5.8%)	-	5 (29.4%)	-	-	-	-	1 (0.4%)	-
Proteinuria	9 (3.4%)	1 (3.0%)	-	-	-	-	-	-	1 (0.5%)	-	-
Eczema/dermatitis	3 (1.1%)	3 (9.0%)	13 (25.5%)	-	-	-	-	-	2 (0.9%)	5 (2.1%)	-
Mucositis oral	3 (1.1%)	3 (9.0%)	5 (9.8%)	-	-	-	1 (1.2%)	7 (38.8%)	-	-	-
Arthralgia	2 (1.0%)	1 (3.0%)	-	-	-	2 (18.0%)	-	-	-	-	1 (9.0%)
Non-tumour bleeding	4 (1.5%)	-	-	-	-	-	-	-	-	1 (0.4%)	
Vomiting	11 (4.1%)	3 (9.0%)	1 (1.9%)	-	-	1 (9.0%)	1 (1.2%)	1 (5.5%)	6 (2.9%)	1 (0.4%)	-
Skin ulceration	5 (1.9%)	-	-	-		-	-		-	-	-
Aspartate aminotransferase increased	8 (3.0%)	1 (3.0%)	12 (23.5%)	-	1 (5.9%)	-	-	2 (11.0%)	-	11 (4.7%)	1 (9.0%)
Alkaline phosphatase increased	-	-	2 (3.9%)	-	-	-	-	-	-	-	1 (9.0%)
Amylase/lipase elevated	-	-	-	-	-	-	-	-	-	21 (9.1%)	-
Hypothyroidism	7 (2.6%)	-	-	-	-	-	-	-	-	3 (1.3%)	-
Hypophysitis	-	-	-	-	-	-	-	-	-	2 (0.8%)	-
Dyspnoea	7 (2.6%)	-	-	-	-	-	-	-	-	4 (1.7%)	-
Tracheal fistula	4 (1.5%)	-	-	-	-	-	-	-	-	2 (0.8%)	-
Fistula	2 (0.8%)	1 (3.0%)	1 (1.9%)	-	-	-	-	-	-	3 (1.3%)	
Aspiration pneumonia/ pneumonitis	6 (2.2%)	1 (3.0%)	3 (5.8%)	-	-	-	-	1 (5.5%)	14 (6.7%)	1 (0.4%)	--
Pneumothorax	1 (0.4%)	-	-	-	-	-	-	-	-	3 (1.3%)	-
Pancreatitis	1 (0.4%)	-	-	-	-	-	-	-	-	-	-
ECG QT prolonged	3 (1.1%)	-	-	-	-	-	6 (7.4%)	-	-	-	-
Acute heart failure	1 (0.4%)	-	-	-	-	1 (9.0%)	1 (1.2%)	-	-	5 (2.1%)	-
Adrenal insufficiency	-	-	-	-	-	-	-	-	-	7 (3.0%)	-
Dose modifications	59 (22.2%)	7 (23.0%)	16 (31.4%)	-	-	-	24 (29.6%)	7 (38.8%)	31 (15.0%)	-	4 (36.4%)
Interruptions	44 (16.5%)	6 (20.0%)	9 (17.6%)	-	-	-	25 (30.8%)	6 (33.3%)	34 (16.5%)	-	-
Withdrawals	17 (6.3%)	1 (3.0%)	12 (23.5%)	-	-	-	12 (14.8%)	1 (5.5%)	9 (4.4%)	5 (2.1%)	-

## Data Availability

Data supporting the results of this study are available within Appendix A and upon request from corresponding author. Raw data used to conduct the analysis is stored and available for validation purposes.

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
