# Peer review of "Modern Therapeutic Approaches in Anaplastic Thyroid Cancer: A Meta-Analytic Review of Randomised and Single Arm Studies on Efficacy and Survival"

_cancers, 2025, doi:10.3390/cancers17050777_

Round 1

Reviewer 1 Report

Comments and Suggestions for Authors

This meta-analysis aims to evaluate the efficacy and safety of modern therapeutic approaches in the treatment of locally advanced, recurrent, and metastatic anaplastic thyroid cancer. The authors conducted a comprehensive literature review, but the flow of the paper requires improvement.

  • Conclusion: The conclusion appears to be overly descriptive and does not align seamlessly with the results and discussion sections. It is recommended to reconsider the conclusion to ensure it appropriately reflects the findings. Introducing subtitles within the results or discussion sections may help establish a clearer narrative leading to the conclusion.
  • Main Goal: What is the primary objective of this meta-analysis for clinical practice? Clearly articulating this goal will help readers understand its significance and relevance.

Minor Points:

  1. Lines 155–156: The authors are advised to use a more standard citation format, such as [30–36], for consistency and readability.

Author Response

Reviewer 1.

comment 1: Main Goal: What is the primary objective of this meta-analysis for clinical practice? Clearly articulating this goal will help readers understand its significance and relevance.

Now, the aim of the present meta-analysis has been rephrased to continue the flow of text, highlighted in red

. In addition, several small studies of IO alone or combined with TKIs have yielded favourable response and progression-free survival (PFS) rates [11].

The present meta-analysis aims to clarify the efficacy and safety of mutational/anti-angiogenesis-directed therapies and contemporary therapies (chemotherapy, radiation therapy, chemoradiation), irrespective of their sequence, to provide some guidance on choosing the most effective regimen for locally advanced, recurrent, and metastatic ATC.

Comment 2: Conclusion: The conclusion appears to be overly descriptive and does not align seamlessly with the results and discussion sections. It is recommended that the conclusion be reconsidered to ensure it appropriately reflects the findings. Introducing subtitles within the results or discussion sections may help establish a clearer narrative leading to the conclusion.

It has completely changed as per the suggestion highlighted in red

While acknowledging the heterogeneity across the studies, this meta-analysis suggests that Dabrafenib-Trametinib and the Lenvatinib-Pembrolizumab combination demonstrate promising efficacy. These treatments were linked to high ORR) and DCR, along with improvements in both OS and PFS. Toxicity profiles were generally manageable. The limited use of NGS and biomarker assays in the included studies was found to be a potential confounding factor. Future clinical trials integrating NGS and biomarker-driven treatment strategies are warranted to explore the outcomes further.

Minor Points:

Comment 3:    Lines 155–156: For consistency and readability, the authors are advised to use a more standard citation format, such as [30–36].

Changed accordingly

Reviewer 2 Report

Comments and Suggestions for Authors

Dear Authors,

Thank you for submitting your comprehensive meta-analysis examining targeted therapy and immunotherapy combinations in anaplastic thyroid cancer (ATC). Your work addresses an important clinical need and provides valuable insights into emerging treatment paradigms. While your manuscript shows promise, I would like to suggest several revisions to enhance its scientific impact and clinical utility.

Introduction: Your historical perspective on doxorubicin therapy is well-noted. However, the introduction would benefit from incorporating more recent developments in chemotherapy. Specifically, I recommend including references and discussion of modern combination regimens, particularly carboplatin/paclitaxel, which has shown meaningful activity in ATC. Additionally, please provide references supporting the prevalence of actionable mutations in ATC, as this forms the foundation for your analysis of targeted therapies.

Results: The presentation of your results is clear and well-organized. The visual representations effectively communicate your findings and facilitate interpretation of the complex dataset.

Discussion: The discussion section requires substantial expansion to fully contextualize your findings within the current treatment landscape. Consider addressing these key areas:

First, please provide a more detailed interpretation of your findings, particularly regarding the effectiveness of different therapeutic combinations and their relationship to molecular profiles. A dedicated section on next-generation sequencing (NGS) findings and their implications for treatment selection in ATC would strengthen the clinical applicability of your work.

Second, the role of PD-L1 expression in ATC requires more nuanced discussion. Please address the controversies surrounding its use as a predictive and prognostic biomarker, including a critical evaluation of potential expression cutoff values and their clinical significance.

Third, while radiotherapy is mentioned in your methods, its integration with systemic therapies deserves more thorough analysis. Please include information about cumulative doses and fractionation schedules. Recent developments in hypofractionated radiotherapy (as highlighted in doi: 10.3390/cancers12092506) suggest promising opportunities for integration with systemic treatments. This warrants detailed discussion, particularly regarding timing and sequencing with targeted and immunotherapy approaches.

These revisions will enhance the manuscript's contribution to the field and provide more practical guidance for clinicians managing ATC. I look forward to reviewing the revised version of your important work.

Author Response

Reviewer 3:

Comment 1; Your historical perspective on doxorubicin therapy is well-noted. However, the introduction would benefit from incorporating more recent developments in chemotherapy. 

Now rephrased and replaced with new reference

Historically, doxorubicin monotherapy or a combination of two or more drugs, such as Paclitaxel, Cisplatin, or carboplatin, Doxorubicin, and Docetaxel, has been considered the most effective treatment. Response rates range from 20% to 50% and significant toxicity [6].

The shift from conventional chemotherapy to precise, molecularly targeted therapies is a rapidly developing phenomenon. Tyrosine kinase inhibitors (TKIs) (Lenvatinib, Sorafenib, Pazopanib) have been trialled since 2008. Lenvatinib has shown meaningful antitumor activity but limited clinical efficacy in ATC [7].

  1. Smallridge RC, Ain KB, Asa SL, Bible KC, Brierley JD, Burman KD, et al; American Thyroid Association Anaplastic Thyroid Cancer Guidelines Taskforce. American Thyroid Association guidelines for management of patients with anaplastic thyroid cancer. Thyroid 2012;22(11):1104-39. doi: 10.1089/thy.2012.0302

Comment 2. Additionally, please provide references supporting the prevalence of actionable mutations in ATC, as this forms the foundation for your analysis of targeted therapies.

Genomic alterations are highly prevalent in ATC compared to differentiated thyroid cancer [8]. Studies indicate that 95.8% of ATC cases harbour at least one genetic alteration in the MAP kinase pathway or the PIK3/Akt pathway [8,9,10]. The most common mutation is BRAF V600E, observed in 8%-87% of cases [8]. Personalized medicine, integrating genomic information, depends on next-generation sequencing (NGS) biomarker assays, which have significantly advanced in ATC. These assays facilitate the identification of targetable mutations and provide timely access to therapeutic agents (BRAF V600E mutations with Dabrafenib plus Trametinib [8]; TRK fusion alterations with Entrectinib or Larotrectinib [9], and RET fusion with Selpercatinib [10]). Despite being based on low-powered studies, these findings have led to the approval of these treatments in ATC.

Despite current recommendations advocating using NGS and biomarker assays, global clinical practice for ATC management remains highly variable. These inconsistencies in implementation and access have highlighted the need for further evaluation, leading to our present meta-analysis.

Comment 3; please provide a more detailed interpretation of your findings, particularly regarding the effectiveness of different therapeutic combinations and their relationship to molecular profiles. A dedicated section on next-generation sequencing (NGS) findings and their implications for treatment selection in ATC would strengthen the clinical applicability of your work.

Locally advanced, recurrent and metastatic ATC remains one of the most challenging malignancies due to its highly aggressive nature and historically limited therapeutic options, which have primarily included systemic chemotherapy and neck irradiation. More recently, there has been a shift towards targeted therapies and IO, offering new avenues for treatment.

This meta-analysis represents the first comprehensive evaluation of various therapeutic approaches in ATC, encompassing mutationally targeted therapies, IO, and novel chemotherapeutic agents to provide insights into therapeutic decision-making within this complex treatment landscape. In contrast, the prior meta-analysis by Huang D et al. focused solely on the efficacy of lenvatinib [7].

The observed ORR of 64.9% in patients receiving dual-targeted therapy (DT) underscores the potent efficacy of BRAF/MEK inhibition, consistent with preclinical and clinical data demonstrating the oncogenic reliance of ATC on the MAPK pathway [36,37]. Similarly, the combination of lenvatinib and pembrolizumab achieved an ORR of 42%, supporting the therapeutic potential of combining angiogenesis inhibition with immune checkpoint blockade in ATC. DCR were also notable, with DT achieving 74.4% and lenvatinib plus pembrolizumab reaching 64.2%, suggesting that durable disease stabilization is possible in patients harbouring actionable mutations. However, despite these encouraging response rates, the OS remains limited at 7.2 months, with the most extended median OS survival observed in DT-treated patients (11.2 months) and those receiving lenvatinib atinib plus pembrolizumab (14.4 months).

Next-Generation Sequencing (NGS) and Molecular-Driven Treatment:

Current guidelines emphasize the routine use of biomarker analysis [61,62], given the high prevalence of genomic alterations in ATC. The most frequently observed mutations include BRAF V600E and MEK alterations (40–50%) [38], TP53 (63%) [55], RET and RAS mutations (22%) [56], and TERT promoter mutations (75%) [62]. Additionally, alterations in PIK3CA (18%) [53], EIF1AX (14%), and PTEN (14%) [63] have been documented, while the prevalence of NTRK fusions and other non-actionable mutations ” such as NESTIN, CCND1, POU5F1, MCL1, MYBL2, IQGAP1, SOX2, and NANOG— remains unknown [64].

In the present meta-analysis, biomarker assessment was reported in only 51.1% of the included studies. Among the patient population, 14.7% were identified as BRAF V600E-mutated, while 9.2% were PD-L1 positive. NTRK1/3 fusions were detected in only 0.7% of cases (12 ATC cases among 83 total NTRK fusion-positive tumours), highlighting a rare but clinically significant subgroup that may benefit from TRK inhibitors. In this meta-analysis, entrectinib demonstrated an ORR of 20%, while larotrectinib achieved an ORR of 29%.

Additionally, the presence of RET fusions in some studies supports the potential role of selective RET inhibitors such as selpercatinib, as evidenced in the LIBRETTO-001 trial [65]. However, this trial was not included in the current meta-analysis, as it only enrolled two ATC patients among 166 individuals with RET-driven tumours. The detection of PIK3CA (1.0%) and RAS mutations (1.1%) suggests alternative pathways for targeted therapy. However, their clinical relevance remains uncertain due to the absence of approved targeted agents for these mutations in ATC. These findings highlight the necessity of incorporating comprehensive genomic profiling into routine clinical practice to optimize treatment selection and improve patient outcomes. The limited availability of molecular data in many studies underscores the need for standardized biomarker assessment in ATC to define treatment strategies better and identify patients most likely to benefit from mutational driven and IO approaches.

  1. Wirth LJ, Sherman E, Robinson B, Solomon B, Kang H, Lorch J, et al. Efficacy of Selpercatinib inRET-Altered Thyroid Cancers. N Engl J Med 2020 A;383(9):825-835. doi: 10.1056/NEJMoa2005651

Comment 4: the role of PD-L1 expression in ATC requires more nuanced discussion. Please address the controversies surrounding its use as a predictive and prognostic biomarker, including a critical evaluation of potential expression cutoff values and their clinical significance.

PD-L1 Expression in ATC:

21.3% of included studies reported PD-L1 expression, with 9.2% of ATC patients identified as PD-L1 positive. This is pretty low, as 22%- 29% of ATC tumour samples have been reported to express PD-L1 [66].

. One of the primary challenges in utilising PD-L1 as a predictive biomarker in ATC is the absence of standardised cutoff values for clinical decision-making. Unlike non-small cell lung cancer (NSCLC) or head and neck squamous cell carcinoma (HNSCC), where PD-L1 expression thresholds (1%-50%) have been validated for pembrolizumab, no established ATC-specific cutoff values currently exist [67].

While our analysis demonstrates that lenvatinib combined with pembrolizumab achieves an ORR of 42% and a median OS of 14.4 months, these findings emphasise further investigation. Future clinical trials should prioritise standardising PD-L1 scoring thresholds and explore potential synergistic treatment strategies, including its combination with angiogenesis inhibitors or RT, to enhance IO efficacy in ATC.

  1. Ahn S, Kim TH, Kim SW, Ki CS, Jang HW, Kim JS, et al. Comprehensive screening for PD-L1 expression in thyroid cancer. Endocr Relat Cancer. 2017 Feb;24(2):97-106. doi: 10.1530/ERC-16-0421
  2. Crosta S, Boldorini R, Bono F, Brambilla V, Dainese E, Fusco N, et al. PD-L1 Testing and Squamous Cell Carcinoma of the Head and Neck: A Multicenter Study on the Diagnostic Reproducibility of Different Protocols. Cancers (Basel). 2021 Jan 14;13(2):292. doi: 10.3390/cancers13020292

Comment 4; Third, while radiotherapy is mentioned in your methods, its integration with systemic therapies deserves more thorough analysis. Please include information about cumulative doses and fractionation schedules. Recent developments in hypofractionated radiotherapy (as highlighted in doi: 10.3390/cancers12092506) suggest promising opportunities for integration with systemic treatments. This warrants detailed discussion, particularly regarding timing and sequencing with targeted and immunotherapy approaches.

Added in results section our findings on RT

RT was administered to 76 out of 980 patients (7.2%). The RT regimen included a daily dose of 2 Gy per fraction over 33 days (totalling 66 Gy) combined with weekly Paclitaxel, with or without Pazopanib, in 71 patients ORR 31%) [30]. Alternatively, a bi-daily fractionation schedule was used in five patients [53], delivering 3.5 Gy per fraction at intervals of more than six hours over two consecutive days (totalling 14 Gy) combined with Pembrolizumab (ORR 40%).

Discussion

Radiation therapy in ATC:

In this meta-analysis, RT was administered only to 7.2% of patients using two fractionation regimens. Most patients (n=71) received conventional fractionation, consisting of a total dose of 66 Gy delivered in 33 fractions (2 Gy per fraction) over 33 days, combined with weekly paclitaxel, with or without pazopanib. In contrast, a smaller cohort (n=5) underwent hypofractionated RT (HypoRT), receiving 14 Gy in 4 fractions (3.5 Gy per fraction) over two consecutive days in combination with pembrolizumab. HypoRT achieved a higher ORR (40%) than conventional RT (31%) [30,53].

This finding aligns with recent work by Oliinyk et al., who evaluated the use of hypoRT in ATC patients treated with 3DRT or IMRT. In their cohort of 17 ATC patients, a cumulative radiation dose of <30 Gy was delivered, with four patients receiving concurrent chemotherapy (carboplatin with paclitaxel or doxorubicin weekly). The median OS was four months (range: 1-12 months), with survival rates of 82%, 55%, and 36% at one, three, and six months, respectively. Subsequent authors performed a systematic review of hypoRT in ATC, supporting its role as an integral component of multimodal treatment, and demonstrating promising clinical outcomes. Given the increasing evidence favouring hypoRT, its incorporation into future clinical trials should be explored, particularly in combination with molecularly targeted agents and IO.

  1. Oliinyk D, Augustin T, Koehler VF, Rauch J, Belka C, Spitzweg C, et al. Hypofractionated Radiotherapy for Anaplastic Thyroid Cancer: Systematic Review and Pooled Analysis. Cancers (Basel) 2020;12(9):2506. doi: 10.3390/cancers12092506

Conclusion has ve totally revised to make flow

This meta-analysis comprehensively evaluated therapeutic strategies for locally advanced, recurrent, and metastatic ATC, focusing on novel mutational-targeted therapy, IO, chemotherapy, and RT.  While dual-targeted therapy (BRAF/MEK inhibition) demonstrated the highest ORR at 64.9%, and lenvatinib plus pembrolizumab achieved a 42% ORR, OS remains a challenge, with median OS reaching only 11.2 months for DT and 14.4 months for lenvatinib plus pembrolizumab.  Despite these improvements in treatment outcomes, our meta-analysis showed underutilization of biomarker-driven treatment, with genomic profiling reported in just over half of the included studies (51.1%), highlighting the critical need for standardized molecular testing.  The limited data on PD-L1 expression (9.2% of patients) and the absence of validated cutoffs in ATC further complicate patient selection for IO.  HypoRT showed promise with a higher ORR (40%) than conventional RT (31%), suggesting its potential within multimodal approaches.  While combination therapy appeared superior as compared to historical TKI monotherapy data,  toxicity profiles varied across treatments, necessitating careful monitoring.  These findings underscore the urgent need for well-designed prospective trials investigating optimal sequencing and combinations of DT, IO, and RT, incorporating standardized molecular profiling (including NGS) to guide treatment decisions, validating PD-L1 cutoffs for IO, and further evaluating the role of HypoRT in combination with systemic therapies.  Such trials should also prioritize evaluating toxicity management strategies and exploring novel mutational-driven and targeted therapies to improve long-term outcomes for ATC patients

Reviewer 3 Report

Comments and Suggestions for Authors

The authors in this paper describes the meta-analysis in targeted therapy in locally advanced, recurrent and metastatic anaplastic thyroid cancer (ATC). The meta-analysis involved and approved 47 studies for a total of 980 patients.

The Focus reported the results of Overall Response Rate (ORR 29.7%), Overall Survival (OS of 7.2 months) and pooled median progression-free survival (PFS 5.4 months).

The results showed that the Dabrafenib/Trametinib (DT) protocol with or without Pembrolizumab and the Lenvatinib plus Pembrolizumab (LP) protocol have superior ORR, PFS and OS.

The conclusions demonstrate the evidence of a first-line DT treatment for BRAFV600-mutated ATC, while the LP protocol seems more effective for BRAFV600 wild-type and PDL1-overexpressing cases.

The conclusions are that the effective associations remain BRAF/MEK inhibitors (DT) with or without immunotherapeutics and protocols involving Tyrosine kinase inhibitors (TKI) (levatinib) associated with immunotherapy (pembrolizumab).

The paper also provides an important meta-analytic evaluation of the toxicity of the various drugs.

Certainly the positive aspects of the meta-analysis are the wide selection of the literature ensured by rational and balanced inclusion and exclusion criteria. This resulted in a large number of patients enrolled (980) which determines the solidity of the analysis and the strength of the sample.

The limitations, also expressed by the Authors, concern:

1. the limitation of randomized trials which associated with retrospective study methods can provide numerous biases and cause a certain generalization;

2. the heterogeneity of studies (prospective and retrospective) which can significantly influence OS.

A further reflection, however, which is very important and to be underlined, is the limited percentage (only 51%) of studies that have performed biomarkers that is even more reduced in BRAFV600 (14.7%) and PDL-1 (9.2%) mutations.

I believe that this point is the true and necessary rationale for the use of targeted therapy and must be highly highlighted by the Authors. In fact, there can be no targeted therapy without monitoring biological markers.

Author Response

Reviewe 3:

Comment 1 and 2: the limitation of randomized trials associated with retrospective study methods can provide numerous biases and cause a certain generalization;
2. the heterogeneity of studies (prospective and retrospective) which can significantly influence OS.

Agree and these points have been acknowledged in the limitation of meta-analysis

Although IPW and standard effect size were used to reduce bias and enable comparisons, the limited number of included randomized studies and variability in study designs among the included studies may have influenced the OS and PFS outcomes..

Round 2

Reviewer 1 Report

Comments and Suggestions for Authors

The authors made sufficient revision.

Reviewer 2 Report

Comments and Suggestions for Authors

Dear authors,

thank you for an interesting systematic review and meta-analysis. All suggestions and issues have been adressed accordingly. The manuscript is highly contributing to the field.